# SHMT2 Induces Stemness and Progression of Head and Neck Cancer

**DOI:** 10.3390/ijms23179714

**Published:** 2022-08-26

**Authors:** Yanli Jin, Seung-Nam Jung, Mi Ae Lim, Chan Oh, Yudan Piao, Hae Jong Kim, QuocKhanh Nguyena, Yea Eun Kang, Jae Won Chang, Ho-Ryun Won, Bon Seok Koo

**Affiliations:** 1Department of Medical Science, College of Medicine, Chungnam National University, Daejeon 35015, Korea; 2Department of Otolaryngology—Head and Neck Surgery, College of Medicine, Chungnam National University, Daejeon 35015, Korea; 3Division of Endocrinology and Metabolism, Department of Internal Medicine, College of Medicine, Chungnam National University, Daejeon 35015, Korea

**Keywords:** head and neck cancer, *SHMT2*, cancer stemness, progression

## Abstract

Various enzymes in the one-carbon metabolic pathway are closely related to the development of tumors, and they can all be potential targets for cancer therapy. Serine hydroxymethyltransferase2 (*SHMT2*), a key metabolic enzyme, is very important for the proliferation and growth of cancer cells. However, the function and mechanism of *SHMT2* in head and neck cancer (HNC) are not clear. An analysis of The Cancer Genome Atlas (TCGA) data showed that the expression of *SHMT2* was higher in tumor tissue than in normal tissue, and its expression was significantly associated with male sex, aggressive histological grade, lymph node metastasis, distant metastasis, advanced TNM stage, and lymphovascular invasion in HNC. *SHMT2* knockdown in FADU and SNU1041 cell lines significantly inhibited cell proliferation, colony formation, migration, and invasion. Additionally, Gene Ontology (GO) and Kyoto Encyclopedia of Genes and Genomes (KEGG) pathway enrichment analyses using TCGA data revealed that *SHMT2* was closely related to cancer stem cell regulation and maintenance. Furthermore, we found that silencing *SHMT2* inhibited the expression of stemness markers and tumor spheroid formation compared with a control group. On the contrary, stemness markers were significantly increased after *SHMT2* overexpression in HEP-2 cells. Interestingly, we found that knocking down *SHMT2* reduced the expression of genes related to the Notch and Wnt pathways. Finally, silencing *SHMT2* significantly reduced tumor growth and decreased stemness markers in a xenograft model. Taken together, our study suggests that targeting *SHMT2* may play an important role in inhibiting HNC progression.

## 1. Introduction

Head and neck cancer (HNC) is the sixth most common cancer worldwide. It is associated with excessive use of alcohol and tobacco [1]. Despite rapid advances in treatment strategies, there is no significant change in overall 5-year survival in patients with HNC [2]. Mortality in HNC patients mainly occurs due to the recurrence of tumors that are resistant to various treatments [3]. Therefore, it is important to better understand the biological mechanisms of HNC progression.

A tumor contains a very small amount of cells that possess self-renewal, pluripotency, and the potential for cancer initiation [4]. These have been considered characteristics of cancer stem cells (CSCs). The properties of CSCs allow them to survive cancer chemotherapy and radiation therapy, and make them prone to causing local or distant relapse [5]. It has been reported that CSCs are correlated with advanced T stage, metastasis, radiation failure, and shortened disease-free survival in HNC [6]. New insights into the role of CSCs in HNC pathobiology could have profound implications for the removal of CSCs to treat these malignancies.

The metabolism of cancer cells is altered in order to sustain their rapid proliferation, migration, and differentiation [7]. In recent years, metabolomics research has emerged as an important framework for exploring the mechanisms of tumor occurrence and development [8,9,10]. One-carbon metabolic units can maintain cellular redox homeostasis and epigenetic stability by sensing nutrients such as glucose and amino acids inside and outside cells, and regulating the synthesis of nucleic acids, proteins, and lipids [11]. Excessive activation of this pathway in cancer provides a favorable environment for tumor progression. In fact, folate antagonists, such as methotrexate and pemetrexed, have constituted the main class of cancer chemotherapy drugs for a variety of cancers, including acute lymphocytic leukemia, lymphoma, and lung cancer, for decades [12,13,14,15]. However, drug resistance is a common problem, and the presence of CSCs is reported to be the main reason for drug resistance in cancer [16,17]. Serine hydroxymethyltransferase-2 (*SHMT2*) is a key enzyme in one-carbon metabolism [18]. It converts serine to glycine, which provides the methyl group of the one-carbon pool required for DNA methylation and nucleotide biosynthesis [19]. Previous studies found *SHMT2* to be overexpressed in various cancers, including B-cell lymphoma, thyroid cancer, and colorectal cancer, and it played an oncogenic role in these cancers [7,19,20]. However, the underlying mechanisms regulating cancer progression still need to be explored.

In the current study, we showed that *SHMT2* is closely related to cancer cell stemness by performing Gene Ontology (GO) and Kyoto Encyclopedia of Genes and Genomes (KEGG) pathway enrichment analyses in The Cancer Genome Atlas (TCGA) dataset for HNC. Next, we focused on the correlation between *SHMT2* and cancer cell stemness in both in vitro and in vivo experiments, and demonstrated for the first time that *SHMT2* promotes the progression of HNC by promoting cancer cell stemness. Therefore, we suggest that *SHMT2* may be a promising target in HNC.

## 2. Results

### 2.1. SHMT2 Was Overexpressed and Associated with Tumor Progression in HNC

As shown in Figure 1A, *SHMT2* expression levels were higher in the HNC samples than in normal tissue in the TCGA database. We next evaluated the correlations between *SHMT2* expression and clinicopathological factors affecting the prognoses of HNC patients. Using the top and bottom 20% of *SHMT2* expression, respectively, the HNC patients with high *SHMT2* expression had a poorer survival rate than those with low *SHMT2* expression (Figure 1B). As shown in Figure 1C–H, male sex (*p* = 0.0026), aggressive histological grade (*p* = 0.0003), lymph node metastasis (*p* = 0.0012), distant metastasis (*p* = 0.0225), advanced AJCC stage (*p* = 0.0016) and lymphovascular invasion (*p* = 0.0012) were significantly associated with high *SHMT2* expression. To explore the expression of *SHMT2* in HNC patients from CNUH (Chungnam National University Hospital), three pairs of patients’ tissues derived from the CNUH data were used for immunohistochemical analysis. As shown in Figure 1I–K, *SHMT2* expression levels were higher in cancer tissues than in their adjacent normal tissues. In addition, considerably higher *SHMT2* protein expression was detected in all four tumor tissues relative to the normal tissues obtained from the same patients (Appendix A). Moreover, RNA sequencing from 4 normal head and neck tissue samples and 43 HNC tissue samples (10 oral cavity, 5 oropharynx, 28 larynx) from the CNUH data was conducted, and we found that the *SHMT2* expression levels were significantly higher in tumor tissues than in normal tissues (Appendix A). Taken together, these data indicated that *SHMT2* could play an oncogenic role in HNC.

### 2.2. SHMT2 Played an Oncogenic Role and Regulated the Epithelial–Mesenchymal Transition in HNC Cells

To explore whether *SHMT2* plays a critical role in HNC cell lines, the expression of *SHMT2* in mRNA and protein were examined in the normal cell lines (hFB) and in seven HNC cell lines (FADU, SNU1041, SNU1076, SCC15, SCC25, HEP-2, YD8). The majority of the HNC cell lines shows significantly higher SHMT2 expression at both mRNA and protein levels (Figure 2A,B). In order to better observe the role of *SHMT2* in HNC, we used shRNA induced knockdown models in two HNC cell lines, FADU and SNU1041 (exhibiting high *SHMT2* expression). In addition, siRNA was used to knock down *SHMT2* expression in the YD8 cell line. *SHMT2* expression was suppressed by both sh*SHMT2*-1 and sh*SHMT2*-2 in both FADU and SNU1041 cells. Cell proliferation was then measured using a WST-1 assay. Depletion of *SHMT2* notably reduced proliferation of FADU and SNU1041 cells (Figure 2C,D). Furthermore, the effects of knocking down *SHMT2* expression on tumor cell clonogenicity were examined. *SHMT2* knockdown also significantly inhibited the colony formation of FADU, SNU1041, and YD8 cells (Figure 2E,F and Appendix A). Next, we investigated the effect of *SHMT2* on cell migration and invasion. After silencing *SHMT2*, FADU, SNU1041, and YD8 cells showed a statistically significant reduction in cell migration and invasion compared with the control group (Figure 2G,H and Appendix A). A correlation between the epithelial–mesenchymal transition (EMT) and cell invasion has been demonstrated in cancer progression. To investigate whether *SHMT2* correlated with the EMT, we examined EMT-related markers, including N-cadherin, E-cadherin, Snail, and Slug, by Western blot analysis. *SHMT2* knockdown caused a significant reduction in the levels of N-cadherin, Snail, and Slug, and an increase in the level of E-cadherin in FADU and SNU1041 cells (Figure 2I,J). Consistently, the expression of N-cadherin was decreased, and the expression of E-cadherin was increased in the SHMT2 knockdown group compared with the control group in YD8 cells (Appendix A). Contrary to the results of silencing *SHMT2*, overexpression of *SHMT2* significantly increased the proliferation, colony formation, migration, and invasion of HEP-2 cells (Appendix A). Overexpression of *SHMT2* also caused a significant decrease in E-cadherin and an increase in N-cadherin. (Appendix A). These data suggested that *SHMT2* plays a critical role in HNC progression.

### 2.3. SHMT2 Was Associated with Cancer Cell Stemness in HNC

TCGA data were processed using R software (version 3.4.3; https://cran.r-project.org/ accessed on 1 May 2021). DEGs between the top and bottom 20% of *SHMT2* expression were identified. Based on a cutoff value of |log2Fold Change| > 1 and *p* < 0.01, there were 3539 DEGs, including 1650 up-regulated genes and 1889 down-regulated genes. Of note, among the up-regulated genes, cell stemness, or cell differentiation-related genes, such as *FGF* family genes (including *FGF3* and *FGF19*), *SOX* family genes (including *SOX2* and *SOX14*), and *ALDH1A1*, were significantly up-regulated DEGs, as labeled in a volcano plot (Figure 3A). GO enrichment analysis was performed using R software, with *p* < 0.05 as the threshold for significance. The significantly enriched GO terms of up-regulated DEGs were closely associated with regulation of the cell cycle, pyrimidine- and purine-containing compound metabolic processes, and one-carbon metabolism biosynthetic processes, which provide an energy source for the rapid and unrestricted growth of cancer cells. Interestingly, in addition to these biological processes, gene sets related to the regulation of stem cell population maintenance and stem cell proliferation were obviously enriched in the up-regulated DEGs (Figure 3B). KEGG analysis was used to identify relevant pathways for the genes. The results showed that the up-regulated DEGs were enriched in the Notch and Wnt signaling pathways, which are well-known stemness pathways (Figure 3C). Furthermore, a list of 34 genes representing the stem cell maintenance gene set from the Molecular Signatures Database (MSigDB) was significantly up-regulated in the patients with high *SHMT2* expression from the TCGA database (Figure 3D). Pearson correlation coefficients were calculated to verify whether *SHMT2* expression was indeed positively correlated with the expression of stemness-related genes. As shown in Figure 3E–H, we found significant positive correlations between *SHMT2* and well-known markers of stemness (*OCT4*, *SOX2*, *ALDH1A1*), as well as the cell differentiation-related *FGF* family gene *FGF19*, which was mentioned above. We also investigated the relationship between cell cycle-related genes (*CCND2*, *CCNE1*, *CDK2* and *CDK4*) and *SHMT2* in HNC. We found positive correlations between *SHMT2* and cell cycle-related genes in the TCGA database (Appendix A). We further validated these results by analyzing the dataset of patients from CNUH. DEGs were identified between the high and low expression of *SHMT2* from 43 HNC tissue samples from CNUH patients, and we performed GSEA using the GO database. As shown in Appendix A, the above-mentioned gene sets related to cancer progression, including cell growth, cell migration, and DNA repair, were enriched in the patients with high *SHMT2* expression from the CNUH dataset. More importantly, the stem cell maintenance, Notch, and Wnt signaling pathways were enriched in the patients with high *SHMT2* expression from the CNUH dataset (Appendix A). These data further suggested that *SHMT2* expression is positively associated with cancer cell stemness in HNC.

### 2.4. SHMT2 Modulated Stemness Properties in HNC Cells

To elucidate the potential variation in cancer cell stemness markers after knockdown of *SHMT2* in HNC cells, shNC, shSHMT2-1, and shSHMT2-2 cells were plated on 6-well plates. mRNA and protein were then collected to perform reverse-transcription quantitative PCR and Western blots. *SHMT2* silencing decreased the mRNA expression of OCT4, SOX2, and NANOG (Figure 4A,B), as well as the corresponding protein expression levels (Figure 4C,D). The same results were observed in YD8 cells silenced with SHMT2-siRNA (Appendix A). The tumor sphere formation assay can be used to identify cells with stem-like characteristics [21]. We collected shNC-, sh*SHMT2*-1-, and sh*SHMT2*-2-transfected FADU and SNU1041 cells after 10 days of tumor sphere assays. Western blots were performed to examine the changes in the protein levels of OCT4, SOX2, and NANOG in FADU sphere-forming cells. Compared to control cells, the sh*SHMT2* populations produced spheroids in smaller numbers and sizes in vitro, indicating their reduced spheroid formation potential (Figure 4E–J). Moreover, silencing *SHMT2* in FADU sphere-forming cells led to significantly lower protein expression of OCT4, SOX2, and NANOG than in the control group (Figure 4K). After transfection with a pCMV6 SHMT2-myc-DDK tagged plasmid, HEP-2 cells were collected to perform reverse-transcription quantitative PCR and Western blots. As shown in Appendix A, SHMT2 overexpression increased the mRNA and protein expression of OCT4, SOX2, and NANOG.

### 2.5. SHMT2 Promoted Stem Cell-like Properties by Activating the Notch and Wnt Signaling Pathways in HNC

To further explore the mechanisms through which *SHMT2* regulates HNC stemness, we performed GSEA using the REACTOME database to analyze the correlations of *SHMT2* expression with the Notch and Wnt signaling pathways. As shown in Figure 5A–F, three gene sets related to Notch and three gene sets related to the Wnt signaling pathway were enriched in the patients with high *SHMT2* expression from the TCGA database. To further demonstrate the effect of *SHMT2* on the Notch and Wnt signaling pathway, we measured the mRNA levels of Notch and Wnt signaling pathway-associated genes in FADU and SNU1041 cells. *SHMT2* knockdown significantly decreased the levels of Notch signaling pathway-related genes, including *NOTCH1*, *NOTCH2*, *NOTCH3*, *JAG2*, *HEY1*, and *HES1*, and Wnt signaling pathway-related genes, including *WNT1*, *WNT3*, *FZD2*, *MYC*, and *CYCLIN D1* (Figure 5G–J). Western blot analysis further confirmed that SHMT2 knockdown decreased the level of Notch signaling pathway proteins (NICD1, HES1) and Wnt/β-catenin signaling pathway proteins (total-β-catenin, active-β-catenin, and CYCLIN D1) in FADU, SNU1041, and YD8 cells (Figure 5K–L and Appendix A).

### 2.6. Depletion of SHMT2 Inhibited Tumor Growth in Mouse Xenograft Models

To confirm the effects of *SHMT2* on HNC cells in an established xenograft mouse model, bioluminescence imaging was performed to observe the tumor cell growth in vivo. The fluorescence area of the groups that were injected with sh*SHMT2* FADU-luc cells was significantly reduced compared to the control group (Figure 6A,B). After sacrificing the mice, tumor volume and weight were measured immediately. The tumor volume and weight were much lower in the groups with tumors induced by shSHMT2-transfected FADU-luc cells (sh*SHMT2*-1 and sh*SHMT2*-2) than in the control group (Figure 6C–E). Western blotting confirmed that the protein expression of *SHMT2* was efficiently silenced in tumors injected with sh*SHMT2* stable cells. Silencing *SHMT2* significantly reduced the expression of stemness markers (OCT4, SOX2, and NANOG). Furthermore, expression of the EMT-related marker N-cadherin was decreased in *SHMT2*-deficient tumors, whereas E-cadherin expression was increased (Figure 6F). Immunohistochemistry also showed lower expression levels of stemness markers, Notch-related, and Wnt-related genes, including HES1 and CYCLIN D1, in the SHMT2-deficient tumor groups than in the control tumor group (Figure 6G).

## 3. Discussion

Despite recent advances in treatment technology, the survival rate for HNC patients remains roughly 50% [22]. The carcinogenesis of HNC involves numerous molecular events, such as the sequential activation of oncogenes and the inactivation of tumor suppressor genes [23]. Cetuximab, a monoclonal antibody against the epidermal growth factor receptor, has been approved as a targeted therapy for HNC. However, this treatment is not suitable for all HNC patients, and once resistance develops, few alternative target drugs are suitable for them [24]. Therefore, development of more treatment options is necessary to improve HNC outcomes.

Cancer-related metabolism has recently become a hot research topic. Several important patterns have been identified, such as altered glucose and glutamine metabolism. Importantly, altered cancer metabolism links various biochemical pathways into a fine-tuned metabolic network in such a way as to maintain the high proliferation rate and rapid progression of malignant cells. Among the key components of cancer metabolism, one-carbon metabolism provides basic needs for anabolic reactions [25]. One-carbon metabolism, which contributes to many biosynthetic pathways that fuel growth, is comprised of a complex network of enzymes [26]. As a key enzyme in one-carbon metabolism, *SHMT2* converts serine into glycine and a one-carbon unit that can participate in nucleic acid biosynthetic processes, including purine and thymidine synthesis [27,28]. Similar results were reflected in the TCGA data that we analyzed. As shown in Figure 3B,C, purine-containing compound metabolic processes and one-carbon metabolism biosynthetic processes were positively enriched in patients with high *SHMT2* expression. At the same time, DNA replication and cell cycle-related gene sets were also positively enriched in patients with high *SHMT2* expression. It was reported that *SHMT2* knockdown inhibited the proliferation and metastasis of colorectal cancer both in vitro and in vivo [7]. In addition, many researchers have also proposed that *SHMT2* overexpression predicts a poor prognosis and promotes progression in various cancers, including bladder cancer, oral cancer, kidney cancer, and lymphoma [29,30,31,32]. Likewise, in this study, we demonstrated that the expression of *SHMT2* in HNC tumor tissues and cell lines was elevated, and its overexpression was also positively related to the histological grade, lymph node metastasis, distant metastasis, AJCC stage, and lymphovascular invasion. Kaplan–Meier survival analysis showed that overexpression of *SHMT2* was associated with significantly lower survival outcomes in HNC patients than in those with low *SHMT2* expression, indicating that high *SHMT2* expression could predict a poor prognosis. We further confirmed that SHMT2 can promote the proliferation, migration, and invasion of HNC cells. These results suggest that *SHMT2* is a key factor that regulates HNC progression.

Researchers have provided various insights into how *SHMT2* is involved in cancer progression. Ye et al. proposed that MYC induces the transcriptional amplification of *SHMT2*, resulting in increased production of NADPH from NADP+, restrained cellular reactive oxygen species, and enhanced cancer cell survival [33]. Furthermore, previous study revealed a potential novel role for SHMT2 as a downstream element of the STAT3 signaling pathway, and this molecular mechanism may be responsible for the transition of prostate cancer towards a more aggressive phenotype [34]. Additionally, other researchers found that knocking down *SHMT2* inhibited cancer progression by inducing the prolongation of the G1 phase of the cell cycle in tongue squamous cell carcinoma [35]. Interestingly, a report also showed that *SHMT2* drove the progression of colorectal cancer by inhibiting the degradation of β-catenin to promote Wnt/β-catenin signaling in a non-metabolic manner [7]. In our study, we screened DEGs between high and low *SHMT2* expression by analyzing HNC sequencing data in the TCGA database, and concluded that *SHMT2* was related to cancer stem cell maintenance and proliferation. Furthermore, we performed an analysis to investigate the relationship between *SHMT2* and stemness markers in the TCGA database. We found significant positive correlations between *SHMT2* and the expression of well-known stemness markers including *OCT4*, *SOX2*, *ALDH1A1*, and *FGF* family members, which showed high fold changes as DEGs between samples with high or low levels of *SHMT2* expression. These results were also confirmed in both in vitro and in vivo experiments in HNC cells. Notch and Wnt are well-known signaling pathways that play important roles in cancer cell differentiation and are responsible for stem cell maintenance. Through the KEGG pathway analysis of DEGs from samples with high and low *SHMT2* expression, we found that *SHMT2* overexpression was closely associated with the Notch and Wnt signaling pathways. We consistently validated these results in FADU cells. The same results were also verified in the CNUH dataset.

Cancer stem cells possess stemness properties, which are reflected in their ability to self-renew and differentiate, contributing to tumor heterogeneity [36]. There is no doubt that these properties may be involved in promoting cancer progression, tumor therapy resistance, and recurrence [37]. It is urgently necessary to continue to gain a deeper understanding of the roles and mechanisms of CSCs in HNC development so that CSC-specific targeting strategies can be developed. One-carbon metabolism processes are critical for the metabolic rewiring of cancers so that they can survive the attack by drugs and adapt to rapid proliferation [38]. Among the enzymes involved in one-carbon metabolism, *SHMT2* has been reported several times to be related to chemotherapy resistance [39,40,41]. In this study, we present for the first time positive correlations of *SHMT2* expression with the expression of cancer stem cell markers, such as *OCT4*, *SOX2*, and *NANOG*, and the aggressiveness of HNC. These results suggest that SHMT2 may serve as an important target for gene therapy that is aimed at two sets of cancer-promoting factors: cancer metabolism and cancer stem cells.

A limitation of this study is that we have not yet demonstrated whether *SHMT2* affects cancer cell stemness and activation of the Notch and Wnt signaling pathways through a metabolic function or a non-metabolic function. Furthermore, the Wnt and Notch signaling pathways are both complex processes. Further in-depth research on the mechanism of how *SHMT2* is involved in Wnt and Notch signaling will continue to be explored in further studies. Another limitation of this study is that we used only one cell line for in vivo experiments. Well defined in vitro models can partially substitute the use of animals in some circumstances. Three-dimensional culture models, including spheroids, a 3D organotypic co-culture (3D-OTC) model, and a 3D collagen-based scaffold model, represent valuable research resources in this field [42,43,44]. In our study, we used a tumor sphere formation assay to identify stem-like characteristics of CSC in the head and neck (Figure 4E–J). This 3D culture model we used could complement the inadequacy of our animal experiment.

In conclusion, our study demonstrated a novel mechanism by which *SHMT2* could influence HNC progression by participating in cancer cell stemness, and our findings suggest that *SHMT2* may serve as an important target in HNC.

## 4. Materials and Methods

### 4.1. Differentially Expressed Gene Identification in TCGA Database

The TCGA database and patients’ data from Chungnam National University Hospital (CNUH) were processed using R software (https://cran.r-project.org/ accessed on 1 May 2021). Differentially expressed genes (DEGs) in the TCGA database were identified among those in the top and bottom 20% of *SHMT2* expression, and DEGs were defined as genes with |log2Fold Change| values > 1 and *p*-values < 0.01. By using the ggplot2 package (http://ggplot2.org accessed on 1 May 2021) in R, a volcano plot was applied for assessing the overall differential expression of genes.

### 4.2. Functional and Pathway Enrichment Analyses of DEGs

Gene set enrichment analysis (GSEA) was performed using the gene sets in the database (https://www.gsea-msigdb.org accessed on 1 June 2021). GO (Gene Ontology) was used to identify enriched functions of genes in biological processes. The KEGG (Kyoto Encyclopedia of Genes and Genomes) was used to identify relevant pathways for the genes. GO and KEGG signaling pathway analyses of the DEGs were performed using R software, with *p* < 0.05 as the threshold for significance.

### 4.3. Cell Lines and Materials

The normal human cell line hFB and the HNC cell lines FADU, SNU1041, SNU1076, SCC15, SCC25, HEP2, and YD8 were from KCLB (Korean Cell Line Bank, Seoul, South Korea). FADU cells were cultured in high-glucose DMEM (Gibco, Grand Island, NY, USA); SNU1076, SNU1041, and YD8 were maintained in RPMI-1640 (Welgene, Gyeongsan, Korea); and SCC15 and SCC25 were cultured in DMEM/F12 (Welgene). HEP-2 cells were cultured in EMEM (ATCC, Manassas, VA, USA). All the cell culture media was supplemented with 10% fetal bovine serum (FBS) and 5% penicillin-streptomycin (Gibco). Cells were grown in a cell culture incubator (37 °C with 5% CO_2_ under humidified conditions).

### 4.4. RNA Isolation and Reverse Transcription-Polymerase Chain Reaction

RNA was extracted from cells using TRIzol (Invitrogen, Carlsbad, CA, USA) and cDNA was synthesized with 2 µg of total RNA and TOPscriptTMRT DryMIX (Enzynomics Inc., Daejeon, Korea), following the manufacturer’s instructions. Amplification was performed using SYBR Green qPCR Master Mix (Thermo Fisher Scientific, Waltham, MA, USA). The polymerase chain reaction (PCR) reactions were performed for 40 cycles of 95 °C for 15 s, 60 °C for 1 min, and 72 °C for 1 min. The primer sequences were as follows: *SHMT2*-F: 5′-TGGAGTAAATTG AGCTGCTG-3′/ *SHMT2*-R: CTGGACATTGACTCCCCACT-3′, *GAPDH*-F: 5′-ACCCAGAAGACTGTG GATGG-3′/ *GAPDH*-R: 5′-TTCTAGACGCAGGTCAGGT-3′, *NOTCH1*-F: 5′-ACTGTGAGGACCTGGTGGAC-3′/ *NOTCH1*-R: 5′TTGTAGGTGTTGGGGAGGTC-3′, *NOTCH2*-F: 5′-AAGCAGAGTCCCAGTGCCTA-3′/ *NOTCH2*-R: 5′-CAGGGGGCACTGACAGTAAT-3′, *NOTCH3*-F: 5′-TGTGGACGAGTGCTCTATCG-3′/ *NOTCH3*-R: 5′-AATGTCCACCTCGCAATAGG-3′, *JAG2*-F: 5′-GCCATGAGAACATTGACGAC-3′/ *JAG2*-R: 5′-GTCGCACGCACAGTAGAAGT-3′, *HEY1*-F: 5′-CGAGGTGGAGAAGGAGAGTG-3′/ *HEY1*-R: 5′-CTGGGTACCAGCCTTCTCAG-3′, *HES1*-F: 5′-GGACGTTCTGGAAATGACA-3′/ *HES1*-R: 5′-CATTGATCTGGGTCATGCAG-3′, *WNT1*-F: 5′-GATTTTGGTCGCCTCTTTGG-3′/ *WNT1*-R: 5′-CGTGGCATTTGCACTCTTG-3′, *WNT3*-F: 5′-CCCGCTCAGCTATGAACAAG-3′/ *WNT3*-R: 5′-ACTTTAGGTGCATGTGGTCC-3′, *CYCLIN D1*-F: 5′-GCATCTACACCGACAACTCCA-3′/ *CYCLIN D1*-R: 5′-GGCACAGAGGGCAACGA-3′, *FZD2*-F: 5′-GCCATCCTATCTCAGCACA-3′/ *FZD2*-R: 5′-CAAGTACGTGGTGACAGTGA-3′. We imported the Ct values provided by the real-time PCR instrument into Excel and used the 2 − ΔΔCt model for relative quantification of real-time quantitative PCR fold changes.

### 4.5. Western Blot Analysis

Each sample was lysed in buffer containing 0.1% sodium dodecyl sulfate, 1.0% Nonidet-P40, 0.5% sodium deoxycholate, 50 mM Tris, pH 8.0, 150 mM NaCl, and a protease inhibitor (Roche Applied Science, Vienna, Austria, pH 7.4). Tissue samples were minced with scissors, and then the total protein was extracted. Electrophoresis was performed as described previously [45]. The primary antibodies involved included anti-*SHMT2*, anti-*OCT4*, anti-*SOX2*, anti-*NANOG*, anti-non-phospho-β-catenin, anti-β-catenin, anti-CYCLIN D1, anti-Slug, anti-Snail, anti-β-actin (1:1000; Cell Signaling Technology Inc., Danvers, MA, USA), anti-GAPDH, anti-E-cadherin, anti-N-cadherin, anti-NICD1, and anti-HES1 (1:1000; Santa Cruz Biotechnology, Dallas, TX, USA). After incubation with the corresponding horseradish peroxidase-conjugated secondary antibodies (1:5000; Santa Cruz Biotechnology), immune reactive bands were visualized by enhanced chemiluminescence detection (Bio-Rad Laboratories, Inc., Hercules, CA, USA).

### 4.6. Short-Hairpin RNA (shRNA) Transfection

*SHMT2* expression was silenced using a commercial pGFP-C-shLenti vector (catalog No: TL309453V, OriGene Technologies, Inc., Rockville, MD, USA). This vector was designed to specifically inhibit SHMT2 expression (shSHMT2-1:5′-AAG ACTGCCAAGCTCCAGGATTTCAAATC-3′; shSHMT2-2:5′- AAGTCAAAGCACACCTGCTGGCAGACATG-3′) and encodes green fluorescent protein as a marker of transfection. Lenti-shRNA scramble control particles were used as a negative control (catalog No: TR30021V, OriGene Technologies, Inc., Rockville, MD, USA). FADU and SNU1041 cells were infected with shRNA-expressing lentiviruses and selected with puromycin. 

### 4.7. SHMT2 Plasmid Transfection

For the experiments of *SHMT2* overexpression, a pCMV6-empty-myc-DDK tagged plasmid (control, PS100001) and a pCMV6 SHMT2-myc-DDK tagged plasmid (*SHMT2,* RC204239) were purchased from OriGene. Then, jetPEI reagent (PolyPlus Transfection, New York, NY, USA) was used to transfect HEP-2 cells according to the manufacturer’s instructions.

### 4.8. Cell Proliferation Assay

In total, 5 × 10^3^ cells per well in 96-well plates were seeded in DMEM. Cell viability was then measured using WST-1 (Roche Diagnostics, Indianapolis, IN, USA). The conversion of WST-1 to formazan was quantitated at 450 nm using an enzyme-linked immunosorbent assay reader.

### 4.9. Cell Migration and Invasion Assay

Transwell membranes (24-well; Costar, Cambridge, MA, USA) were coated with Matrigel for 6 h for the invasion assay or without Matrigel for the migration assay. In total, 2 × 10^5^ cells in serum-free medium were seeded onto the upper chamber, and 750 μL of medium with 10% FBS were added to the lower chamber. After incubation for 24 h (for migration) and 48 h (for invasion), the cells adhering to the upper surface of the membrane were removed with a cotton swab. The invasion or migration cells that adhered to the lower surface were stained with crystal violet and counted in four representative fields using light microscopy (×40 magnification).

### 4.10. Clonogenic Assay

Cells were seeded in 6-well plates at 1 × 10^3^ cells/well in 2 mL of medium. The medium was replaced with new medium every three days, and the cells allowed to grow for 10 days. The colonies were fixed in 4% formaldehyde, stained with crystal violet, and then imaged and counted.

### 4.11. Tumor Sphere Formation Assays

Tumor sphere formation was induced using plates with low attachment. Cells were seeded at 2 × 10^4^ cells per well. Media were prepared using recombinant human EGF, human FGF, B27 supplement, and 1% penicillin/streptomycin in corresponding media, and changed every 2–3 days. Tumorsphere formation was quantified 10 days after initial seeding using ImageJ. Spheres with a diameter of at least 50 μm were deemed tumor spheres. The collected spheroid cells were used for Western blot analysis.

### 4.12. Animal Experiments

Six-week-old BALB/c nude mice were obtained from Orient Bio (Seongnam, South Korea). The animals were housed at 24 °C with a 12-h day/night cycle under specific pathogen-free conditions. They had ad libitum access to a gamma-ray-irradiated laboratory rodent diet (Purina Korea). All experiments were performed in accordance with the relevant guidelines and regulations of the animal care unit at Chungnam National University. The animal protocols for these experiments were approved by the Ethics Committee of Animal Experimentation of Chungnam National University (No. CNUH-020-A0036-1). Fifteen mice were randomly divided into three groups. Control-transfected Luciferase-expressing FADU (FADU-Luc) cells (shNC) and shRNA-transfected FADU-Luc cells (shSHMT2-1 and shSHMT2-2) were prepared, and 2 × 10^6^ cells in 200 μL of suspension mixture with Matrigel were subcutaneously inoculated into the lower left flank of the mice. The tumor dimensions were measured using a caliper, and tumor volumes were estimated as follows: tumor volume = length × width^2^ × 0.52, where length represents the largest tumor diameter and width represents the diameter perpendicular to the length. The tumors were harvested and used for histological analyses. All the animal experiments were repeated at least twice, with similar results.

### 4.13. In Vivo Imaging

Bioluminescence imaging was performed using an in vivo imaging system consisting of a Lumina XRMS instrument (PerkinElmer, Waltham, MA, USA). Animals were injected intraperitoneally with 150 mg/kg D-luciferin (Promega, Madison, WI, USA). Images were analyzed with the Living Image software (Caliper Life Sciences, Waltham, MA, USA).

### 4.14. Histological and Immunohistochemical Analysis

Tissue samples were fixed in 4% formalin solution and paraffin-embedded. For hematoxylin and eosin (H&E) staining, tissue sections were deparaffinized in xylene, hydrated in graded alcohol solutions, and stained with H&E. The samples were examined under an automatic digital slide scanner (Pannoramic MIDI; 3DHISTECH, Budapest, Hungary) after mounting. For immunohistochemistry, tissue sections were deparaffinized in xylene, hydrated in graded alcohol solutions, and heated (100 °C) for 15 min in Antigen Retrieval Citra Solution (pH 6.0) for antigen retrieval. For single immunostaining, endogenous peroxidase activity was blocked in a 1% hydrogen peroxide solution (Sigma-Aldrich, St. Louis, MO, USA) in phosphate-buffered saline (PBS) with 0.3% Triton X-100 for 30 min at room temperature. The sections were incubated with the indicated antibodies overnight at 4 °C, and then incubated with the corresponding horseradish peroxidase-conjugated secondary antibody. Finally, 3,3′-diaminobenzidine (DAB; Dako, Agilent, Santa Clara, CA, USA) was used to detect these labeled antibodies and the nucleus was stained with hematoxylin. After rinsing with PBS, the samples were mounted and analyzed using an automatic digital slide scanner (Pannoramic MIDI; 3DHISTECH).

### 4.15. Statistical Analysis

All in vitro experiments were repeated three times, and statistical significance was analyzed using the Student’s *t*-test. The in vivo results were analyzed using one-way analysis of variance. Data were presented as means ± standard deviation. To evaluate associations between SHMT2 and stemness marker expression in The Cancer Genome Atlas (TCGA) data, we used Pearson correlation analyses. A *p*-value < 0.05 was considered to indicate statistical significance. The statistical analyses were performed using SPSS version 26.0 (IBM Corp., Armonk, NY, USA).

## Figures and Tables

**Figure 1 ijms-23-09714-f001:**
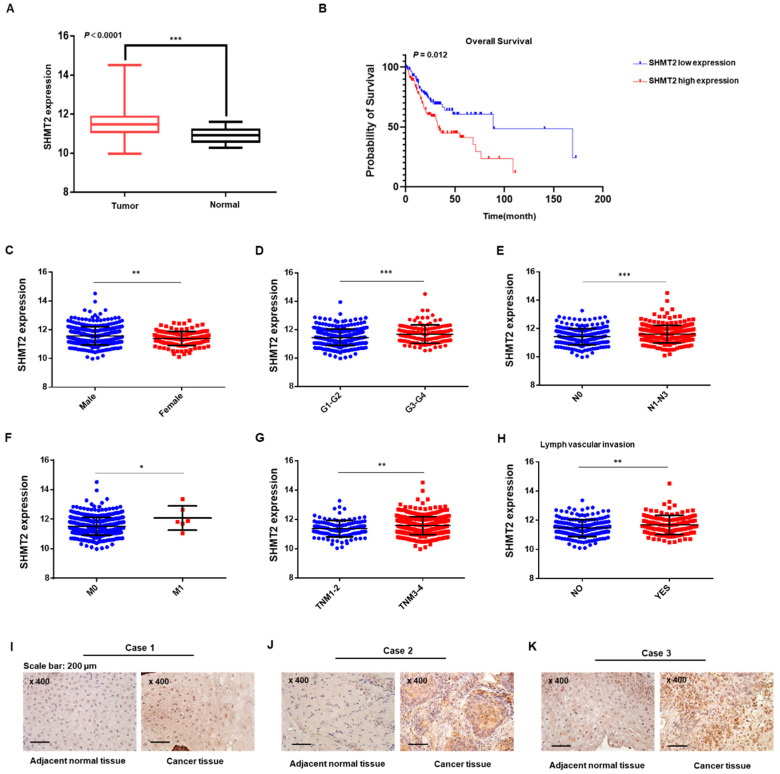
***SHMT2* was overexpressed and associated with tumor progression in HNC.** (**A**) *SHMT2* was expressed significantly more highly in tumor tissues than in normal tissues of HNC in The Cancer Genome Atlas (TCGA) database (*p* < 0.0001). (**B**) Patients from TCGA with low *SHMT2* expression had a better survival rate than those with high *SHMT2* expression (*p* = 0.012). (**C**–**H**) Expression of *SHMT2* in HNC based on the TCGA database according to sex, histological grade (**G**), lymph node metastasis (**N**), distant metastasis (**M**), TNM (primary tumor, lymph node, distant metastasis) stage, and lymphovascular invasion. (**I**–**K**). The *SHMT2* levels of three HNC tissue samples and their adjacent normal tissue samples were examined by immunohistochemistry. *p* < 0.05 was recognized as statistically significant (* *p* < 0.05, ** *p* < 0.01, *** *p* < 0.001).

**Figure 2 ijms-23-09714-f002:**
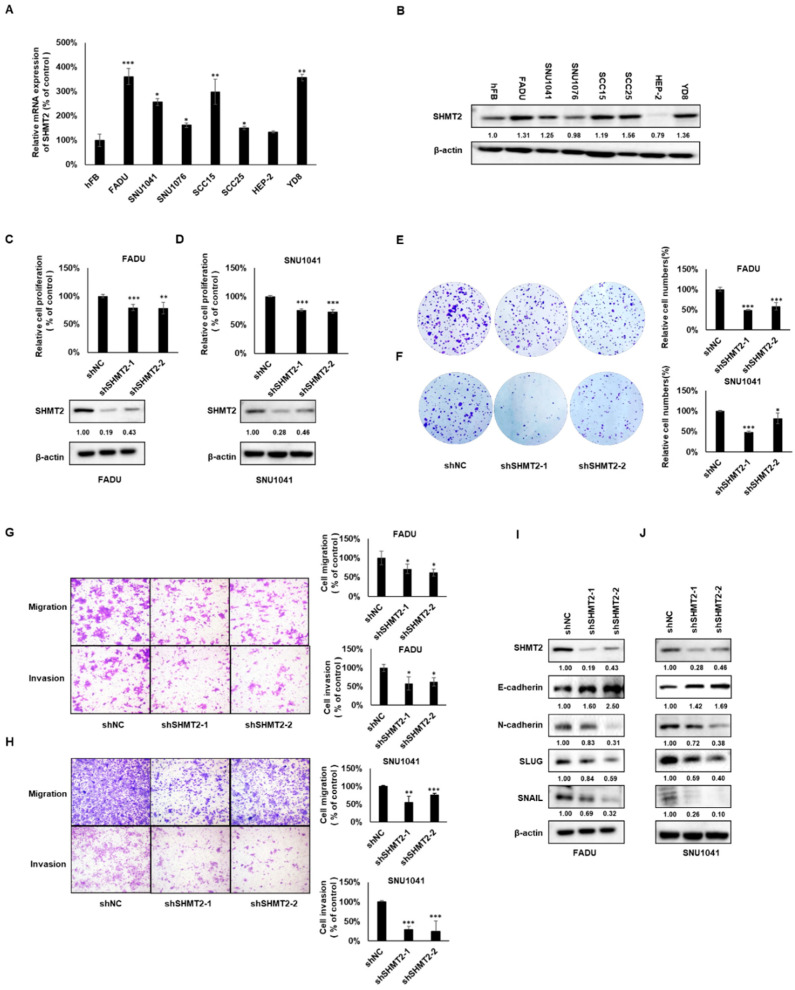
***SHMT2* played an oncogenic role and regulated the epithelial–mesenchymal transition (EMT) in HNC cells.** (**A**,**B**) Normal head and neck cell lines (hFB) and HNC cell lines (FADU, SNU1041, SNU1076, SCC15, SCC25, HEP-2, and YD8) were subjected to reverse-transcription PCR analysis and Western blot analysis. (**C**,**D**) FADU and SNU1041 cells were infected with scramble-sh (shNC) or *SHMT2* knockdown (shSHMT2-1, shSHMT2-2) lentivirus for 2 weeks. The stable cell lines were selected with puromycin (1 mg/mL), and cell viability was analyzed using the WST-1 assay. The levels of SHMT2 were detected by Western blots. (**E**,**F**) Representative images of colony formation assays from FADU and SNU1041 stable cell lines and quantitative analysis of colony formation assays. (**G**,**H**) FADU and SNU1041 cells transfected with shSHMT2-1 and shSHMT2-2 displayed significantly lower migration and invasion capacity than those transfected with each negative control (shNC). (**I**,**J**) Differences in the expression of EMT markers among FADU and SNU1041 cells transfected with shSHMT2 or shNC were detected by Western blotting. Data were presented as mean ± SD of three independent experiments. Differences were considered relevant at *p* < 0.05 (* *p* < 0.05, ** *p* < 0.01, *** *p* < 0.001).

**Figure 3 ijms-23-09714-f003:**
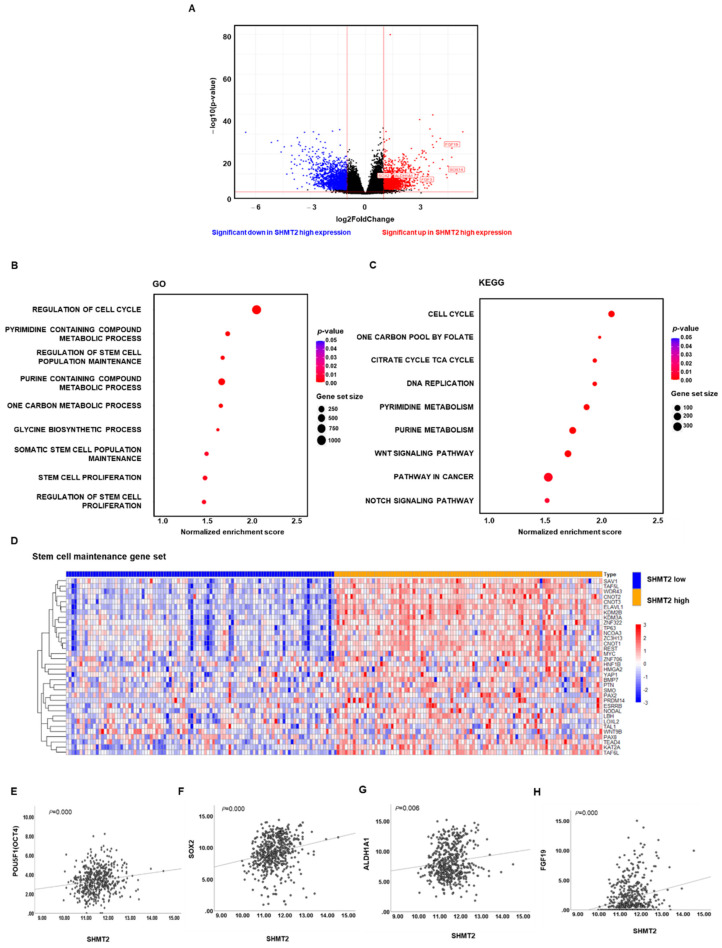
**GO and KEGG enrichment analyses of DEGs between the low and high *SHMT2* in the TCGA HNC cohort.** Genes with |log2Fold Change|> 1 and a *p*-value < 0.01 were defined as differentially expressed genes (DEGs). The up-regulated genes are marked in red, and the down-regulated genes are marked in blue. (**A**) Volcano plot of DEGs. (**B**,**C**) GO enrichment and KEGG pathway analysis results for DEGs. (**D**) The GSEA heat map shows differences in the expression of the core enrichment genes involved in the stem cell maintenance gene set between the patients with high and low *SHMT2* expression. (**E**) Correlation between *POU5F1* (*OCT4*) and *SHMT2*, (**F**) Correlation between *SOX2* and *SHMT2*, (**G**) Correlation between *ALDH1A1* and *SHMT2*, and (**H**) Correlation between *FGF19* and *SHMT2*. Differences were considered relevant at *p* < 0.05.

**Figure 4 ijms-23-09714-f004:**
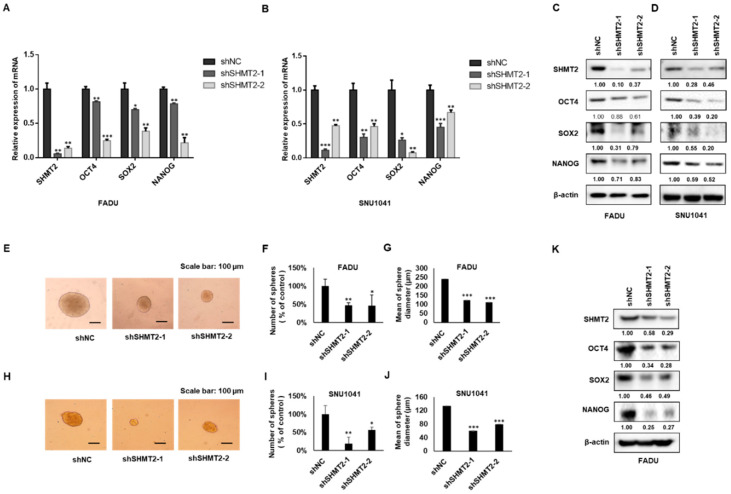
**Silencing *SHMT2* attenuated stemness properties in FADU and SNU1041 cells.** (**A**,**B**) Compared with the control group, *SHMT2* mRNA expression was significantly decreased in the sh*SHMT2*-transfected FADU and SNU1041 cells; the mRNA expression of *OCT4, SOX2*, and *NANOG* was also significantly decreased after silencing SHMT2. (**C**,**D**) The protein expression of OCT4, SOX2, and NANOG was also significantly decreased in sh*SHMT2*-transfected FADU and SNU1041 cells. (**E**) Representative images of tumor spheres in shNC- and shSHMT2-transfected FADU cells (scale bar: 100 μm). (**F**) A comparison of the number of spheres between shNC and sh*SHMT2*-transfected FADU cells. Spheres were counted after 10 days of incubation. The sphere number of each group was normalized to the shNC group. (**G**) Comparison of sphere diameters. (**H**) Representative images of tumor spheres in shNC- and shSHMT2-transfected SNU1041 cells (scale bar: 100 μm). (**I**) Comparison of the number of spheres between the shNC and sh*SHMT2*-transfected SNU1041 cells. The sphere number of each group was normalized to the shNC group. (**J**) Comparison of sphere diameters in SNU1041 cells. (**K**) The protein expression of OCT4, SOX2, and NANOG was significantly decreased in sh*SHMT2*-transfected FADU sphere-forming cells. Differences were considered relevant at *p* < 0.05 (* *p* < 0.05, ** *p* < 0.01, *** *p* < 0.001). All experiments were repeated at least three times.

**Figure 5 ijms-23-09714-f005:**
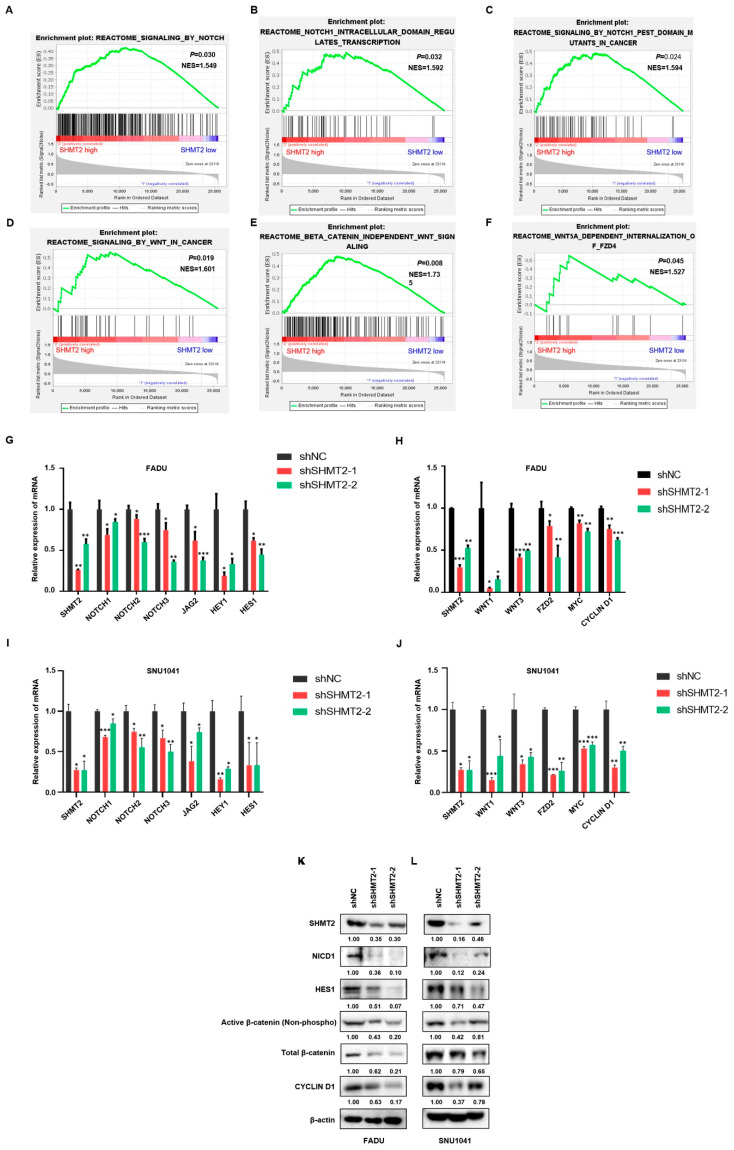
***SHMT2* promoted stem cell-like properties by activating the Notch and Wnt signaling pathways in HNC.** (**A**–**F**) Notch- and Wnt pathway-related gene sets were enriched in HNC patients with high *SHMT2* expression in the TCGA dataset. (**G**–**J**) Reverse-transcription quantitative PCR analysis of Notch and Wnt pathway-related genes with *SHMT2* knockdown in FADU and SNU1041 cells. (**K**,**L**) Western blot analysis of Notch and Wnt pathway-related proteins with SHMT2 knockdown in the FADU and SNU1041 cells. Differences were considered significant at *p* < 0.05 (* *p* < 0.05, ** *p* < 0.01, *** *p* < 0.001).

**Figure 6 ijms-23-09714-f006:**
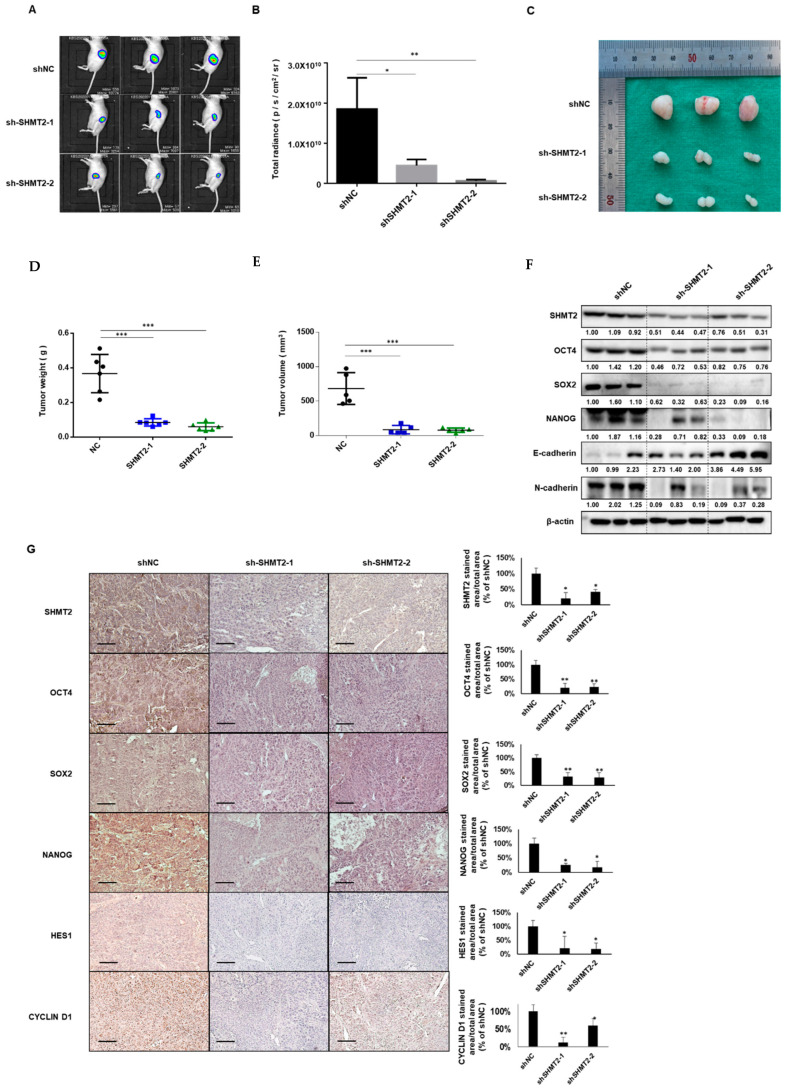
**Tumor growth in a mouse xenograft model after *SHMT2* knockdown.** 2 × 10^6^ control-transfected Luciferase-expressing FADU (FADU-Luc) cells (shNC) and shRNA-transfected FADU-Luc cells (shSHMT2-1 and sh*SHMT2*-2) were subcutaneously injected into 15 BALB/c-nude mice. (**A**,**B**) Representative final tumor images of cancer cells tracked with the in vivo imaging system following the injection of mice with FADU-Luc cells. (**C**–**E**) Images of tumors at the experimental endpoint. The tumors weight and volume were measured at the time of sacrifice. (**F**) Western blots from xenograft tumor tissues. Changes in the protein expression of *SHMT2*, *OCT4*, *SOX2*, *NANOG*, E-cadherin, and N-cadherin in the xenograft tissues of each group. (**G**) Representative images of immunohistochemical staining of the negative control group and sh*SHMT2* cell-injected groups. Scale bar, 100 μm. Results were analyzed using one-way analysis of variance. Differences were considered significant at *p* < 0.05 (* *p* < 0.05, ** *p* < 0.01, *** *p* < 0.001).

## Data Availability

Not applicable.

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
