# Peer review of "SHMT2 Induces Stemness and Progression of Head and Neck Cancer"

_ijms, 2022, doi:10.3390/ijms23179714_

Round 1

Reviewer 1 Report

The aim of this manuscript is to discuss the current understanding of SHMT2, by highlighting the stemness and progression relationship in head and neck cancer, in human and animal models.

Even if the review presents a densely organized structure and is based on well-synthetized data, there are aspects to be mentioned to make the manuscript fully readable. For these reasons, the article requires minor changes.

1. Line 86, in the result section, please provide the full name of CNUH.

2. Figure 2.1, shSHMT2-1 in E-cadherin seems to increase, but the value is 0.59.

3.SHMT2 decreases E-cadherin and increases N-cadherin, and what is the rule and participates what signaling pathways in the regulation of E-cadherin and N-cadherin?

4. In figure 3 B&C, both the GO and KEGG analysis software showed that the cell cycle is the enrichment. Did the authors investigate the relationship between cell cycle genes and SHMT2?

Reviewer 2 Report

The authors of the present work studied the role of SHMT2 in the carcinogenesis processes of head and neck cancers (HNCs). In vitro analysis showed the implication of this gene in cell proliferation, colony formation, migration, and invasion. Moreover, in silico and in vitro experiments demonstrated the association of SHMT2 modulation with stemness processes in HNC cells.

The manuscript is well written but contains significant flaws and, for this reason, the article needs some major revisions.

The manuscript would benefit from the following:

-        The first in vitro experiment was performed using eight cell lines but the authors selected only two of them to continue the study (except for HEP-2 cells in some experiments). This aspect limits the translational value of the data and the reasons of this decision are not described. The authors should discuss the motivations of this experimental choice.

-        The in silico analysis were performed on a large cohort of HNCs, not divided by histotypes. The two cell lines selected for the data confirmation were established from hypopharyngeal tumors and for this reason, the in vitro results can be associated to all kinds of HNCs. The authors should perform the same experiments on cell lines derived by other histotypes too or report this limitation in the discussion.

-        Another limitation of the present study is the low number of in vivo analysis performed (it has been used only one cell line). The murine model represents an expensive approach that takes long time. Well defined preclinical models are needed to partially substitute the use of animals. As performed by the authors, 3D models represent valuable research resources in this field.  For this reason, the authors should underline this aspect through a short overview of the in vitro 3D models. The following references should be included in the manuscript: doi: 10.3390/cancers12082330, doi: 10.20892/j.issn.2095-3941.2020.0482 and doi: 10.3389/fonc.2022.960340.

-        Study limitation should be reported.

Author Response

We appreciate the constructive comments of all reviewers and have striven for sincere responses about all the reviewer’s concerns.

Reviewer 2

1. The first in vitro experiment was performed using eight cell lines but the authors selected only two of them to continue the study (except for HEP-2 cells in some experiments). This aspect limits the translational value of the data and the reasons of this decision are not described. The authors should discuss the motivations of this experimental choice.

Response

Thank you for valuable comments. Considering the experiments we are going to perform, we selected two cell lines with higher SHMT2 expression than normal cell for the further study. And we thought that using two cell lines was sufficient to validate our research. But now we agree with the reviewer’s point. As you mentioned in reviewer’s comment 2, we also think that the choice of cell lines derived from same part of the head and neck tissue have some limitations for this study. Therefore, we selected YD8 cell line (established from oral cancer) with relatively high SHMT2 expression and performed similar experiments to improve the rationality of the experiment. In addition, we added brief discussion of this issue in Result 2 (page3, line112-115).

2. The in silico analysis were performed on a large cohort of HNCs, not divided by histotypes. The two cell lines selected for the data confirmation were established from hypopharyngeal tumors and for this reason, the in vitro results can be associated to all kinds of HNCs. The authors should perform the same experiments on cell lines derived by other histotypes too or report this limitation in the discussion.

Response

Thank you for your considerate comment. We agree with the reviewer that cell lines derived from same head and neck area could be the limitation of this article. Following your suggestion, we have added a cell line derived from oral cancer (YD8) and performed the same experiments on YD8 cell line and added the data in Supplementary Fig. 2.

We added the results in Result 2 (page4, line120-121/ page4, line123-124/ page4, line129-132), Result 4 (page8, line206-207), and Result 5 (page9, line250-251).

3. Another limitation of the present study is the low number of in vivoanalysis performed (it has been used only one cell line). The murine model represents an expensive approach that takes long time. Well defined preclinical models are needed to partially substitute the use of animals. As performed by the authors, 3D models represent valuable research resources in this field. For this reason, the authors should underline this aspect through a short overview of the in vitro3D models. The following references should be included in the manuscript: doi: 10.3390/cancers12082330, doi: 10.20892/j.issn.2095-3941.2020.0482 and doi: 10.3389/fonc.2022.960340.

Study limitation should be reported.

Response

Thank you for your valuable comment. Based on your suggestion, we have written this limitation into the discussion as below and cited the references you mentioned above (page14, line 368-375):

‘Another limitation of this study is that we used only one cell line for in vivo experiments. Well defined in vitro models can partially substitute the use of animals in some circumstances. 3D culture models including spheroids, a three-dimensional organotypic co-culture (3D-OTC) model, and 3D collagen-based scaffold model represent valuable research resources in this field [42-44]. In our study, we used tumor sphere formation assay to identify stem-like characteristics of CSC in head and neck (Fig 4E-J). This 3D culture model we used could complement the inadequacy of our animal experiment.’

Round 2

Reviewer 2 Report

Now the manuscript is sitable for the publication